# Using Nano-Luciferase Binary (NanoBiT) Technology to Assess the Interaction Between Viral Spike Protein and Angiotensin-Converting Enzyme II by Aptamers

**DOI:** 10.3390/biotech14010020

**Published:** 2025-03-15

**Authors:** Meng-Wei Lin, Cheng-Han Lin, Hua-Hsin Chiang, Irwin A. Quintela, Vivian C. H. Wu, Chih-Sheng Lin

**Affiliations:** 1Department of Biological Science and Technology, National Yang Ming Chiao Tung University, Hsinchu 30068, Taiwan; eva1001cat@gmail.com (M.-W.L.); a0975273923@gmail.com (C.-H.L.); chhxzk0314@gmail.com (H.-H.C.); 2Produce Safety and Microbiology Research Unit, United States Department of Agriculture, Agricultural Research Service, Albany, CA 94710, USA; irwin.quintela@usda.gov; 3Center for Intelligent Drug Systems and Smart Bio-devices (IDS2B), National Yang Ming Chiao Tung University, Hsinchu 30068, Taiwan

**Keywords:** severe acute respiratory syndrome coronavirus 2 (SARS-CoV-2), spike protein, nano-luciferase binary technology (NanoBiT), aptamer, angiotensin-converting enzyme type II (ACE2)

## Abstract

Nano-luciferase binary technology (NanoBiT)-based pseudoviral sensors are innovative tools for monitoring viral infection dynamics. Severe acute respiratory syndrome coronavirus 2 (SARS-CoV-2) infects host cells via its trimeric surface spike protein, which binds to the human angiotensin-converting enzyme II (hACE2) receptor. This interaction is crucial for viral entry and serves as a key target for therapeutic interventions against coronavirus disease 2019 (COVID-19). Aptamers, short single-stranded DNA (ssDNA) or RNA molecules, are highly specific, high-affinity biorecognition elements for detecting infective pathogens. Despite their potential, optimizing viral infection assays using traditional protein–protein interaction (PPI) methods often face challenges in optimizing viral infection assays. In this study, we selected and evaluated aptamers for their ability to interact with viral proteins, enabling the dynamic visualization of infection progression. The NanoBiT-based pseudoviral sensor demonstrated a rapid increase in luminescence within 3 h, offering a real-time measure of viral infection. A comparison of detection technologies, including green fluorescent protein (GFP), luciferase, and NanoBiT technologies for detecting PPI between the pseudoviral spike protein and hACE2, highlighted NanoBiT’s superior sensitivity and performance, particularly in aptamer selection. This bioluminescent system provides a robust, sensitive, and early-stage quantitative approach to studying viral infection dynamics.

## 1. Introduction

The coronavirus disease 2019 (COVID-19) pandemic, caused by severe acute respiratory syndrome coronavirus 2 (SARS-CoV-2), drove significant advancements in diagnostic and therapeutic strategies, particularly for the rapid and precise detection of viral components and interactions [1]. Understanding protein–protein interactions (PPIs), particularly those involving viral proteins and their host counterparts, is essential for developing effective countermeasures [2,3]. Among the emerging tools, nano-luciferase binary technology (NanoBiT) offers a robust platform for studying and detecting PPI [4], surpassing traditional approaches in sensitivity and dynamic range [5,6]. In this study, we focused on the utility of NanoBiT technology in detecting PPI during SARS-CoV-2 infection, comparing its advantages with other fluorescence- and luminescence-based pseudovirus systems, such as green fluorescent protein (GFP) and luciferase assays, and leveraging NanoBiT for the selection of aptamers against the viral infection.

NanoBiT technology represents a significant advancement in bioluminescent detection, offering a highly sensitive, low-background, and scalable platform for PPI studies [7,8]. The NanoBiT system relies on a split luciferase design comprising a large subunit (LgBiT) and a small subunit (SmBiT) (Figure 1). These subunits are attached to two proteins of interest, and their interaction brings the subunits together, reconstituting luciferase activity and generating a measurable luminescent signal. NanoBiT overcomes many shortcomings of GFP and traditional luciferase assays, providing higher sensitivity, real-time monitoring capabilities, and an extended dynamic range [9]. These attributes make it particularly well-suited for studying weak or transient PPIs, such as those involving viral proteins.

Aptamers, single-stranded DNA (ssDNA) or RNA molecules selected for their ability to bind specific targets with high affinity, are valuable tools for detecting and modulating PPI [10,11]. Furthermore, aptamers have been successfully utilized in therapeutic and diagnostic applications for various pathogens, including parasites, bacteria, and viruses. These attributes position aptamers as a promising tool for targeting SARS-CoV-2 and its variants [12,13,14]. Numerous researchers have identified aptamer sequences specific to SARS-CoV-2 through in silico analysis or GFP- and luciferase-based selection methods [15,16,17,18,19]. However, there remains a need for more sensitive and accurate biosensors to comprehensively validate these aptamers for SARS-CoV-2 applications.

In the present study, we aim to compare the performance of GFP, luciferase, and NanoBiT technologies in detecting the PPI between the pseudoviral spike and cellular angiotensin-converting enzyme II (ACE2) and highlight the advantages of NanoBiT in aptamer selection. We also evaluate the comparative advantages of NanoBiT over GFP and luciferase technologies, emphasizing its application in aptamer selection and validation for SARS-CoV-2 diagnosis and therapeutic interventions.

## 2. Materials and Methods

### 2.1. Cell Culture

Human embryonic kidney 293T cells (HEK293T; ATCC CRL-3216, Manassas, VA, USA) and HE293T cells with high human ACE2 expression (HEK293T-hACE2; ALSTEM, Richmond, CA, USA) were cultured in high-glucose Dulbecco’s modified Eagle medium (DMEM) supplemented with 10% fetal bovine serum (FBS), incubated at 37 °C in a humidified atmosphere with 5% CO_2_, and generally subcultured every 2–3 days.

### 2.2. Generation of the HEK293T/LgBiT-hACE2 Cell Line

To establish the HEK293T/LgBiT-hACE2 cell line, we followed the methodology described in our previous study [8]. The hACE2 gene was amplified from the hACE2 cDNAs of K-18 hACE2 mice (GenBank: NM_021804.1) using PCR with specific primer pairs. The resulting LgBiT-hACE2 PCR fragment was digested with *Nhe I* and *Hpa I* and then subcloned into the appropriate vector. The integrity of the DNA constructs was confirmed via restriction enzyme digestion, and all constructs were sequenced to ensure accuracy. The constructed LgBiT-hACE2 plasmid DNA was transduced into HEK293T cells, and then, the cell clone, HEK293T/LgBiT-hACE2, that was able to stably express the LgBiT-hACE2 fusion protein was screened.

### 2.3. Production of the NanoBiT-Based Pseudovirus

A lentivirus packaging system was used to produce the NanoBiT-based pseudovirus [8]. Three vectors were chosen for construction: pCMVΔR8.91, pLAS2.1w.PeGFP-I2-Puro (RNAiCore, Taipei, Taiwan), and pcDNA3.3_Omicron BA.2-SmBiT. HEK293T was seeded onto a 24-well plate and cultured for 24 h. Thereafter, 15 µg of plasmid DNA (Ration of transfer vector:packaging vector:envelope vector = 10:9:1) and Lipofectamine 2000 Transfection Reagent were co-transfected into the HEK293T cells. The NanoBiT-based pseudovirus (BA.2-SmBiT pseudovirus) was harvested at 72 h after DNA transfection.

### 2.4. NanoBiT Detection

HEK293T/LgBiT-hACE2 cells were seeded onto a 96-well plate at a density of 1 × 10^4^ cells/well and incubated for 24 h. The aptamers were pre-incubated with the pseudovirus at 37 °C for 4 h. Subsequently, the LgBiT-hACE2 cells were infected with the BA.2-SmBiT pseudovirus in the presence of furimazine substrate (Promega, Madison, WI, USA). The luminescence signal from the infected cells was measured using a Synergy HT multimode microplate reader (BioTek Instruments, Winooski, VT, USA) to assess inhibition efficacy.

### 2.5. Production of the GFP- and Luciferase-Based Pseudovirus

The GFP-based pseudovirus was constructed using three vectors: pCMVΔR8.91, pLAS2.1w.PeGFP-I2-Puro (RNAiCore), and pcDNA3.3_SARS2_Omicron BA.2 (Addgene, Watertown, MA, USA). Similarly, the luciferase-based pseudovirus was generated using pCMVΔR8.91, pLAS2w.FLuc.Ppuro (RNAiCore), and pcDNA3.3_SARS2_Omicron BA.2 (Addgene). HEK293T cells were seeded onto a 24-well plate and cultured for 24 h. Plasmid DNA (15 µg total) at a ratio of transfer vector–packaging vector–envelope vector = 10:9:1 was co-transfected with Lipofectamine 2000 Transfection Reagent (Thermo Fisher Scientific, Waltham, MA, USA) into the HEK293T cells. The pseudovirus was harvested 72 h after post-transfection and used for subsequent experiments.

### 2.6. Fluorescence of GFP and Luminescence of Luciferase Detection

The HEK293T-hACE2 cells were infected with GFP- and luciferase-based pseudoviruses for 3, 24, and 48 h at 37 °C. GFP expression was observed through microscopic analysis at 3, 24, and 48 h post-infection= and cells were collected after infection to analyze GFP intensity using a Synergy HT multimode microplate reader (BioTek Instruments). HEK293T-hACE2 cells were seeded onto a 96-well plate at a density of 1 × 10^4^ cells/well and incubated for 24 h. Subsequently, the HEK293T-hACE2 cells were infected with the luciferase-based pseudovirus in the presence of D-luciferin (GoldBio, St. Louis, MO, USA). Luminescence signals from the infected cells were measured using the Synergy HT multimode microplate reader to evaluate inhibition efficacy. Luminescence was further monitored using the Caliper Spectrum IVIS system (Caliper LifeSciences, Hopkinton, MA, USA), and images were acquired with an exposure time of 2 s.

### 2.7. Western Blot Analysis

Cell total lysate was collected in PRO-PREPTM protein extraction solution (INTERCHIM, San Diego, CA, USA). Western blot analysis was performed as previously described [20]. The primary monoclonal antibodies used for LgBiT (Promega), ACE2, viral spike protein, and β-actin (GeneTex, Irvine, CA, USA) were diluted to 1:1000. The secondary antibodies were diluted to 1:10,000. The substrate containing the secondary antibodies was conjugated with horseradish peroxidase (HRP), and the immunoblots were visualized using enhanced chemiluminescence (ECL; GeneTex). The target locations of the bands were detected based on protein molecular weight using an iBright Imaging System (Thermo Fisher Scientific). The expression levels of hACE2 and spike proteins were calculated and compared with those of β-actin for each sample.

### 2.8. Aptamer Sequences and Structure Prediction

The sequence of aptamers used during the study is 5′-ATC CAG AGT GAC GCA GCA TCG AGT GGC TTG TTT GTA ATG TAG GGT TCC GGT CGT GGG TTG GAC ACG GTG GCT TAG T-3′ (76-mer aptamer) [17], 5′-ATC CAG AGT GAC GCA GCA TTT CAT CGG GTC CAA AAG GGG CTG CTC GGG ATT GCG GAT ATG GAC ACG T-3′ (67-mer aptamer) [18], 5′-CGC AGC ACC CAA GAA CAA GGA CTG CTT AGG ATT GCG ATA GGT TCG G-3′ (46-mer aptamer) [19], and Cy5-5′-CGC AGC ACC CAA GAA CAA GGA CTG CTT AGG ATT GCG ATA GGT TCG G-3′ (Cy5-labeled 46-mer aptamer). The aptamers were synthesized by Genomics BioSci & Tech (Taipei, Taiwan). The sequence of a random DNA control is 5′-TGA TTG AGT GAC GCA GCA TGG ACA CGG TGG CAA CAG-3′ [21]. The predicted structure of the aptamers was analyzed on the UNAFold Web Server (https://www.unafold.org/mfold/applications/dna-folding-form.php; accessed on 10 November 2024) [22]. The protein–DNA docking interactions were analyzed using the HDOCK Server (http://hdock.phys.hust.edu.cn/; accessed on 1 December 2024).

### 2.9. Cy5-Aptamer and Confocal Microscopy

Fluorescence inverted microscopy was conducted to confirm the specific binding of aptamers to the pseudoviral spike. SARS-CoV-2 pseudoviral virions were pretreated with 0.1 and 1 µM concentrations of the Cy5-labeled 46-mer aptamer (Genomics BioSci & Tech) for 4 h. Next, 2.5 × 10^5^ HEK293T-hACE2 cells were seeded onto 12 mm coverslips in 24-well plates. Following cell attachment, the medium was removed, and the cells were incubated with 1 µg/mL Hoechst 33342 (Servicebio, Wuhan, China) in medium for 30 min. Thereafter, the medium was removed, and the cells were washed twice with Dulbecco’s phosphate-buffered saline (DPBS; Thermo Fisher Scientific). The cells were then incubated with Cy5-labeled 46-mer-treated virions for 1 h at 37 °C. After incubation, the aptamer solution was removed, and the cells were washed three times with DPBS. The cells were fixed with 300 μL of 4% formaldehyde at room temperature in the dark for 15 min. After washing, excess liquid was removed, and 10 μL of Fluoromount-G^®^ (Southern Biotech, Birmingham, AL, USA) was applied to mount the coverslip. The cells were visualized using a multiphoton and confocal microscope system (Leica, TCS-SP5-X AOBS; Leica, Wetzlar, Germany).

### 2.10. Statistical Analysis

All the data are expressed as the mean ± standard deviation (SD). Statistical significance was assessed using Student’s *t*-test for the experiments. The specific statistical test used for each determination is marked in the corresponding figure legends, and statistically significant differences were noted as * *p* < 0.05, ** *p* < 0.01, and *** *p* < 0.001.

## 3. Results

### 3.1. Validation of HEK293T/LgBiT-hACE2 Cells and the SmBiT-BA.2 Pseudovirus

The structural maps of LgBiT-hACE2 and BA.2-SmBiT plasmids are shown in Figure 2A. To validate the generation of the HEK293T/LgBiT-hACE2 cell line and SmBiT-BA.2 pseudovirus, ACE2 and spike protein levels were assessed using immunoblotting. The HEK293T/LgBiT-hACE2 cells were expected to express both ACE2 (120 kDa) and LgBiT-hACE2 (140 kDa) proteins, respectively. As shown in Figure 2B, both ACE2 and LgBiT-hACE2 were detected in the HEK293T/LgBiT-hACE2 cells; in comparison, LgBiT-hACE2 was not detected in the HEK293T-hACE2 cells. Regarding the SmBiT fragment, due to its small size (11 a.a.), no significant difference in molecular weight was detected between the SmBiT-BA.2 pseudovirus and BA.2 pseudovirus (Figure 2C).

### 3.2. NanoBiT-Based and Luciferase-Based SARS-CoV-2 Spike Pseudovirus Infection

NanoBiT-derived bioluminescence was detected in HEK293/LgBiT-hACE2 cells as early as 3 h post-transduction with the SmBiT-BA.2 pseudovirus. The luminescence signal peaked at 3 h post-infection, reaching a maximum intensity of 45.2 × 10^6^ p/sec/cm^2^. Over time, the luminescence signal exhibited a time-dependent decline, as shown in Figure 3A.

Bioluminescence activity was detected in HEK293-hACE2 cells as early as 3 h after transduction by the luciferase-based SARS-CoV-2 pseudovirus. While almost no luminescence intensity was detected at 3 h post-infection, the intensity of luminescence was 0.37 × 10^6^ p/sec/cm^2^ at 24 h post-transduction. The strongest luminescence intensity, 24.2 × 10^6^ p/sec/cm^2^, could be observed in HEK293-hACE2 cells at 48 h post-transduction (Figure 3B).

### 3.3. GFP-Based Pseudoviral SARS-CoV-2 Infection

GFP expression was detected in HEK293-hACE2 cells as early as 3 h after transduction by the GFP-based SARS-CoV-2 pseudovirus. In comparison, almost no GFP expression was detected at 3 h post-infection (Figure 4A). The significant GFP expression (7.97 × 10^4^ A.U.) was detectable 24 h after pseudovirus infection, and the strongest GFP expression (15.6 × 10^4^ A.U.) could be observed in HEK293-hACE2 cells 48 h post-infection of the pseudovirus (Figure 4B).

### 3.4. Prediction of Aptamer Sequences

The secondary structures of the aptamers and a series of their truncated sequences were simulated using the UNAFold Web Server. As shown in Figure 5, all aptamers exhibited typical stem-loop structures. For the 76-mer aptamer, two possible secondary structures with low Gibbs free energy (ΔG) were predicted. The 67-mer aptamer was associated with three possible secondary structures, one of which had the lowest ΔG of −5.21 kcal/mol. The 46-mer aptamer was predicted to form two distinct secondary structures.

### 3.5. Inhibition of the Aptamers on NanoBiT-Based Pseudoviral SARS-CoV-2

The identified aptamer candidates were evaluated to determine their relative binding affinities using the NanoBiT pseudoviral system. The 76-mer, 67-mer, and 46-mer aptamers significantly inhibited the interaction between the ACE2 receptor of HEK293T/LgBiT-hACE2 cells and the spike protein of the SmBiT-BA.2 pseudovirus. Among all of the aptamers, the 46-mer demonstrated the lowest limit of detection (*p* < 0.05 compared with the intensity of the 76-mer aptamer) for the Omicron BA.2 variant pseudovirus. The 76-mer, 67-mer, and 46-mer aptamers showed significant anti-pseudoviral SARS-CoV-2 ability with half-maximal inhibitory concentration (IC_50_) of 3.61, 1.64 and 0.70 µM, respectively (Figure 6A). The secondary structure of the 46-mer aptamer was predicted, with a ΔG of −4.70 kcal/mol, as shown in Figure 5C. The corresponding tertiary structure of the 46-mer aptamer and SARS-CoV-2 BA.2 variant spike protein was modeled using the HDOCK SERVER. Due to the high binding affinity of the 46-mer aptamer, molecular docking was performed to predict the spatial structure of the protein–ligand complex and identify the critical nucleotides involved in binding. The top three docking sites of the 46-mer aptamer on the spike protein are highlighted in yellow (Model 1), green (Model 2), and red (Model 3) (Figure 6B).

### 3.6. Aptamers as Detection Reagents for SARS-CoV-2

To validate the spike-binding ability of the 46-mer aptamer, which was selected via the NanoBiT system, HEK293T-hACE2 cells, stably overexpressing hACE2, and SARS-CoV-2 pseudovirus were utilized in this study. The SARS-CoV-2 pseudovirus was packaged with spike protein as an envelope protein using a lentivirus packaging system. Its RNA genome, containing the GFP gene, was driven by the CMV promoter, enabling infection of ACE2-expressing cells. Infected cells were identified through GFP fluorescence imaging. The cy5-labeled 46-mer aptamer was incubated with SARS-CoV-2 pseudoviral virions and subsequently added to HEK293T-hACE2 cells. Confocal fluorescence microscopy confirmed that with increasing amounts of the 46-mer aptamer, fewer green fluorescent pseudovirus-infected cells were detected, indicating that pseudovirus entry into cells is contingent upon the 46-mer aptamer binding to the spike protein (Figure 7).

## 4. Discussion

PPI plays a critical role in the life cycle of SARS-CoV-2, particularly in the interaction between the viral spike protein and the ACE2 receptor on the cell membrane, which facilitates viral entry [23,24]. Monitoring these interactions is crucial for understanding the molecular mechanisms of infection and for developing therapeutic interventions [8,25]. Traditional PPI detection methods, such as yeast two-hybrid screens, fluorescence resonance energy transfer (FRET) assays using GFP, and bioluminescence-based luciferase systems, have been widely employed [26,27]. However, each method presents limitations. GFP-based assays often suffer from high background fluorescence, low sensitivity, and photobleaching, while luciferase systems can be limited by signal decay and a narrow dynamic range [28,29,30].

GFP-based pseudoviral assays, although widely utilized for visualizing interactions, require specialized imaging equipment and are susceptible to issues such as autofluorescence and photobleaching, which reduce their sensitivity and limit their applicability for real-time detection [31,32]. In contrast, luciferase-based pseudoviral assays offer improved sensitivity and dynamic range but are prone to signal instability and interference from small-molecule inhibitors [33]. Additionally, both GFP- and luciferase-based systems rely on the expression of their respective markers in host cells post-infection, leading to a time delay in monitoring PPI [27,34]. NanoBiT-based pseudoviral assays overcome these limitations by providing a robust, low-background, and scalable platform that enables real-time quantitative detection of PPI with enhanced sensitivity and reliability (Table 1).

Precise diagnostic or therapeutic targeting of drugs is crucial for effective treatment strategies [35]. Over the past decade, aptamers have emerged as promising tools due to their synthetic flexibility, stability, and ability to be engineered and customized [36,37]. These characteristics make aptamers an attractive alternative to traditional antibodies, particularly in scenarios requiring high stability and low immunogenicity. Numerous researchers have identified aptamer sequences specific to SARS-CoV-2 through in silico analysis or GFP- and luciferase-based selection methods [15,16,17,18,19].

The possible secondary structures of aptamers with low ΔG values were predicted using the UNAFold Web Server, with the 67-mer exhibiting the lowest ΔG of −5.21 kcal/mol. In general, a lower ΔG suggests a more stable secondary structure, which may contribute to the stability of target–aptamer complexes [38,39]. However, it is important to consider whether ΔG is always directly correlated with binding affinity [40]. There remains a need for more sensitive and accurate biosensors to validate these aptamers fully for SARS-CoV-2 applications. In our NanoBiT selection, the 46-mer aptamer, which had a moderate ΔG among the aptamer candidates, demonstrated the strongest inhibitory ability against pseudoviral SARS-CoV-2.

Additionally, the K_D_ values of the 76-mer, 67-mer, and 46-mer aptamers against the SARS-CoV-2 receptor-binding domain (RBD) were 7 nM, 19.9 nM, and 0.13 nM, respectively. The IC_50_ of the 76-mer and 46-mer aptamers for blocking the interaction between RBD and ACE2 receptors were 5 nM and 0.42 nM, respectively [17,18,19]. The smaller the K_D_ value, the stronger the binding affinity of the ligand to the target, and the lower the IC_50_ value, the higher the inhibitory potency [41]. Our present data are consistent with these quantitative results.

To further validate the NanoBiT pseudoviral selection system, the Cy5-labeled 46-mer aptamer was developed and used to examine the cellular localization and distribution of the 46-mer aptamer and pseudoviral SARS-CoV-2. Cy5 conjugates, commonly used with antibodies, peptides, and proteins, are pH-insensitive across a broad pH range (pH 4 to pH 10) [42]. A notable advantage of using long-wavelength dyes such as Cy5 is the minimal autofluorescence of biological specimens in this spectral region [43]. CyDye-labeled aptamers represent a colorimetric detector suitable for diverse analytical applications [44]. The confocal microscopy images confirmed that the entry of pseudoviral SARS-CoV-2 into host cells was disrupted by the binding of the 46-mer aptamer to the spike protein.

Despite these advancements, more sensitive and accurate biosensors are required to fully validate these aptamers for SARS-CoV-2 applications, especially in blocking the interaction between the viral spike protein and cellular ACE2. In this context, NanoBiT has proven valuable for studying the interaction between the spike protein and ACE2, offering a robust platform for screening inhibitors and therapeutic molecules. By conjugating aptamers to the NanoBiT subunits, this system not only facilitates the identification of aptamers with high binding affinity to viral proteins but also enables functional assays to evaluate their efficacy in disrupting PPI. Furthermore, the modular design of NanoBiT supports rapid adaptation to emerging SARS-CoV-2 variants, underscoring its potential as a versatile and sustainable approach for pandemic preparedness.

In summary, through the NanoBiT pseudoviral system, we verified the utility of the 46-mer aptamer, a high-affinity ssDNA aptamer capable of binding to the viral spike protein and inhibiting pseudoviral entry. Our findings suggest that 46-mer aptamer holds promise as a potential therapeutic option for SARS-CoV-2 treatment, providing a novel approach to the prevention and management of COVID-19.

## 5. Conclusions

In the present study, we systematically elucidated the difference among NanoBiT-, GFP-, and luciferase-based pseudoviral systems through a direct comparative analysis. NanoBiT emerged as the superior platform, demonstrating significantly higher luminescence intensity and markedly shorter detection times relative to GFP and luciferase systems. This enhanced performance is attributed to the robust bioluminescent signal generated by the NanoBiT system, which facilitates high sensitivity and rapid assay readouts. Furthermore, integrating NanoBiT with aptamer-based approaches synergistically combines precise detection of PPI with the capability to identify and develop targeted inhibitors for virus infection. This dual functionality not only improves the accuracy and efficiency of molecular interaction studies but also provides a versatile platform for advancing SARS-CoV-2 diagnostics and therapeutic interventions. By addressing the limitations of existing methodologies, NanoBiT offers a transformative tool that enhances the ability to manage current viral challenges and fortify preparedness against future pandemics.

## Figures and Tables

**Figure 1 biotech-14-00020-f001:**
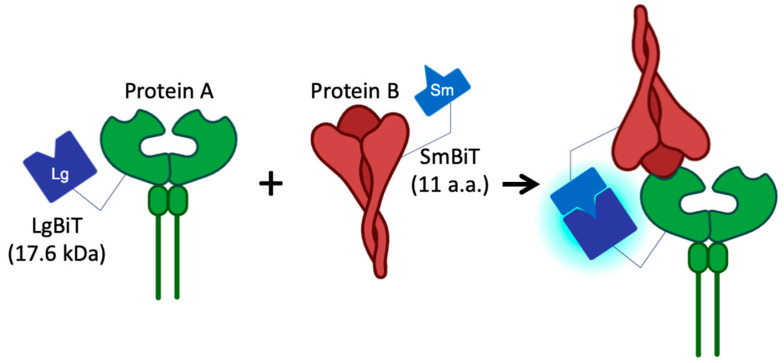
NanoLuc^®^ Binary Technology (NanoBiT) is a structural complementation reporter system composed of a large BiT (LgBiT) subunit and a small complimentary peptide (SmBiT). For the study of protein–protein interaction (PPI), the complimentary peptide is small BiT (SmBiT; 11 amino acid peptides, a.a.), which has been optimized to have low affinity for LgBiT (17.6 kDa). The LgBiT and SmBiT subunits are expressed as fusions to target proteins of interest and are expressed in cells. When the two proteins interact, the subunits come together to form an active enzyme and generate a bright luminescent signal in the presence of substrate.

**Figure 2 biotech-14-00020-f002:**
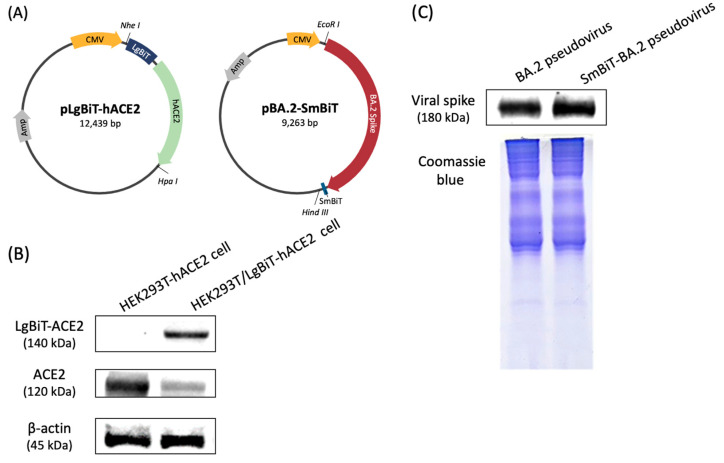
Establishment of the NanoBiT-based pseudoviral system and Western blot analysis of HEK293T/LgBiT-hACE2 cells and SmBiT-BA.2 pseudovirus. (**A**) Plasmids of Lg-hACE2 and BA.2-SmBiT. (**B**) Protein expression levels of LgBiT-hACE2, ACE2, and β-actin in HEK293T-hACE2 and HEK293T/LgBiT-hACE2 cells. (**C**) Viral spike protein in the BA.2 and SmBiT-BA.2 pseudoviruses. β-actin and Coomassie blue staining were used as protein loading controls.

**Figure 3 biotech-14-00020-f003:**
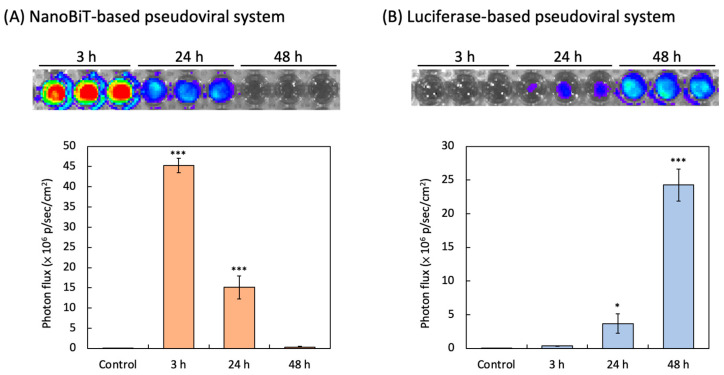
Detections of the luminescent intensity by NanoBiT-based and luciferase-based pseudovirus infection. (**A**) The bioluminescence imaging of HEK293T/LgBiT-hACE2 cells infected with SmBiT-BA.2 pseudovirus by 3 h, 24 h, and 48 h were shown. (**B**) The bioluminescence imaging of HEK293T-hACE2 cells infected with BA.2 pseudovirus (Luc) by 3 h, 24 h and 48 h were shown. The NanoBiT- and luciferase-derived luminescence intensity in HEK293T/LgBiT-hACE2 cells was quantified [total flux photons per second and per area (p/sec/cm^2^) intensity]. Data from all experimental values were shown as the mean ± SD for each group (n = 3). * *p* < 0.05 and *** *p* < 0.001 compared with Control group (blank).

**Figure 4 biotech-14-00020-f004:**
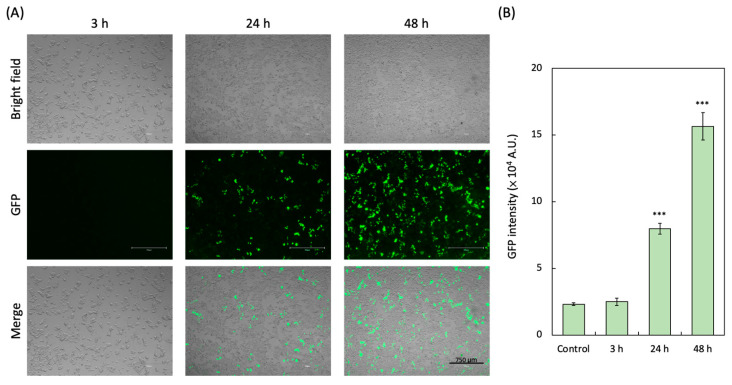
The HEK293T-hACE2 cells were infected by GFP-based SARS-CoV-2 pseudovirus. (**A**) The represented GFP images of HEK293T-hACE2 cells infected with BA.2 pseudovirus (GFP) by 3 h, 24 h, and 48 h were shown. (**B**) The GFP intensity in HEK293T-hACE2 cells was quantified. Data from all experimental values were shown as the mean ± SD for each group (n = 3). *** *p* < 0.001 compared with the Control group.

**Figure 5 biotech-14-00020-f005:**
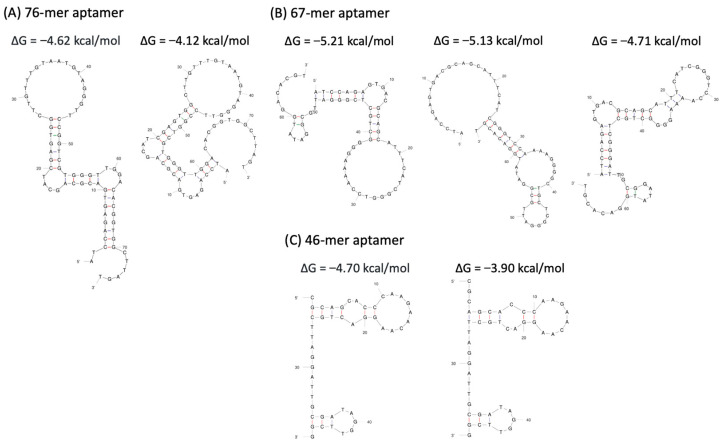
Prediction of the possible secondary structure of the selected aptamers and their free energy. (**A**) 76-mer aptamer; (**B**) 67-mer aptamer; (**C**) 46-mer aptamer. The 76-mer, 67-mer, and 46-mer aptamer presented 2, 3, and 2 typical stem-loop structures and had the lowest ΔG of −4.62, −5.21, and −4.70 kcal/mol, respectively.

**Figure 6 biotech-14-00020-f006:**
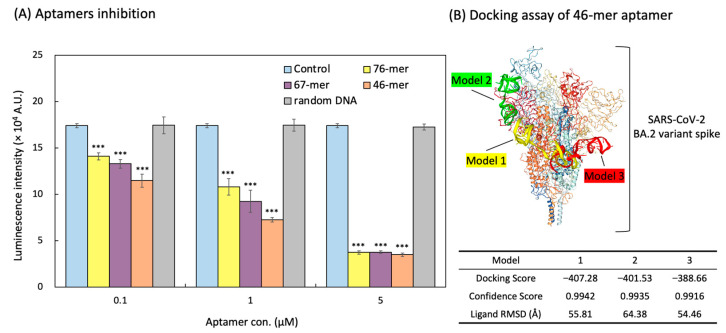
Comprehensive analysis of aptamer performance in NanoBiT-based SARS-CoV-2 pseudoviral detection. (**A**) Inhibition efficiency of the aptamers assessed using the NanoBiT-based pseudoviral detection system. Data from all experimental values were shown as the mean ± SD for each group (n = 3). *** *p* < 0.001 compared with the Control group. (**B**) Tertiary structure of the SARS-CoV-2 BA.2 variant spike protein in complex with the 46-mer aptamer. The three putative binding sites of the 46-mer aptamer in spike protein are highlighted in yellow (Model 1), green (Model 2), and red (Model 3).

**Figure 7 biotech-14-00020-f007:**
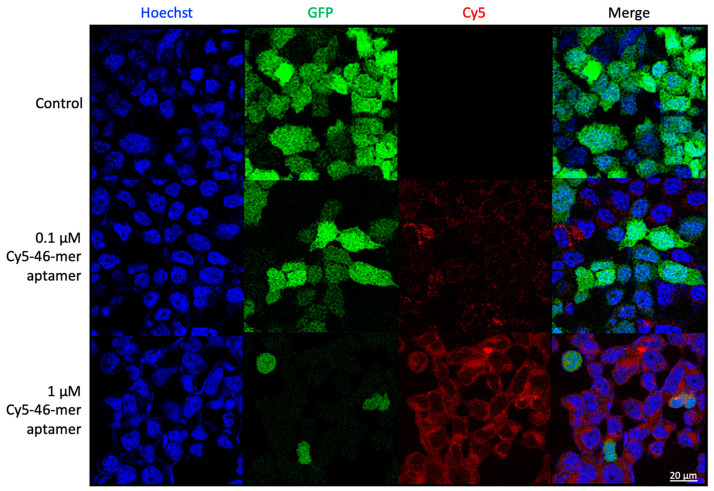
Confocal microscopy images of pseudovirus-infected HEK293T-hACE2 cells transfected with Cy5-46-mer aptamer. Green fluorescence signal was generated by GFP in cells. Red fluorescence signal was generated by Cy5-labeled aptamer. The nuclei were stained blue with Hoechst.

**Table 1 biotech-14-00020-t001:** Comparison among NanoBiT-, GFP- and luciferase-based SARS-CoV-2 spike pseudoviral systems.

Pseudoviral System	Reporter	Activation	Substrate	Optimal Detection Time	Strongest Intensity
**NanoBiT**	NanoBiT luciferase fragments, including pseudovirus-SmBiT and live cell-LgBiT	PPI between pseudovirus and cells	Furimazine	3 h	45.26 × 10^6^ p/sec/cm^2^
**GFP**	GFP pseudovirual plasmid and expressed GFP in host cells	Reporter expression in the infected cell	Not applicable	48 h	15.6 × 10^4^ A.U.
**Luciferase**	Luciferase pseudoviral plasmid and expressed liciferase in host cells	Reporter expression in the infected cell	D-luciferin	48 h	24.26 × 10^6^ p/sec/cm^2^

## Data Availability

The data presented in this study are available upon request to the corresponding author.

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
