# Peer review of "Using Nano-Luciferase Binary (NanoBiT) Technology to Assess the Interaction Between Viral Spike Protein and Angiotensin-Converting Enzyme II by Aptamers"

_biotech, 2025, doi:10.3390/biotech14010020_

Round 1

Reviewer 1 Report

Comments and Suggestions for Authors

Visualization of viral infection dynamics is useful for understanding viral infection mechanisms and developing treatment methods. In the previous study, the authors developed a biosensor that enables visualization of SARS-CoV-2 spike protein and hACE2 receptor interaction at an early stage of pseudoviral infection by using NanoLuc Bionary Technology (DOI: 10.1016/j.bios.2024.116630). Here, Meng-Wei Lin and colleagues applied their system to screen an ssDNA aptamer, which targets omicron BA.2 variant spike protein and inhibits SARS-CoV-2 BA.2. Indeed, the authors have successfully demonstrated that the 46-mer DNA aptamer efficiently inhibited pseudoviral entry. I think the experimental design is appropriate, and the manuscript is well written overall. Therefore, I support the publication in BioTech.

Minor comments:

Page 2, Line 45 – Regarding the NanoBiT, I suggest that the authors cite the original paper (DOI: 10.1021/acschembio.5b00753).

Page 5, Figure 2 – Could the authors provide the uncropped version of the immunoblotting images as a supplementary data?

Page 8, Figure 6A – Could the authors provide the information on ‘random DNA’ used as a negative control, in the ‘Materials and Methods’ section?

Reviewer 2 Report

Comments and Suggestions for Authors

The manuscript titled Using nano-luciferase binary (NanoBiT) technology to assess the interaction between viral spike protein and angiotensin converting enzyme II by aptamers shows the use of NanoBiT system to evaluate the capacity of three different aptamers to inhibit the interaction of SARS-CoV-2 BA.2 spike protein with ACE-2, in comparison with GFP and luciferase transduction systems.

In my opinion the article is so interesting and well written but cannot be accepted in the current version.

Major comments:

-The determination of the inhibition efficiency using the NanoBIT system is the principal experiment of this manuscript. In the form that is shown, do not include quantitave results, EC50 determination, the sequence and size of the random DNA used as control, the conditions of the negative control, etc. It is mandatory that the details can be included to compare with the other techniques and conclude all the advantages for this system.

-The conclusion that the 46-mer aptamer is the best selection can be supported by more quantitative results.

-Include the parameters to define the specificity of the NanoBIT system. To the end all the advantages of this system can not be compared because many information is not shown.

Comments on the Quality of English Language

The article is well written in all the sections.

Round 2

Reviewer 2 Report

Comments and Suggestions for Authors

In the new version of the manuscript, information about the IC50 determination of the aptamers is included by data obtained previously. The principal comment is that the  determination of these values can be obtained using the NanoBIT system to determine if this technique as robust as other techniques. Please include an experiment for complete this requirement.

Round 3

Reviewer 2 Report

Comments and Suggestions for Authors

In my opinion the authors included the requirements demanded, the manuscript can be accepted in the new version.